# Sensitivity of point-of-care testing C reactive protein and procalcitonin to diagnose urinary tract infections in Dutch nursing homes: PROGRESS study protocol

Sacha D Kuil,[1] Soemeja Hidad,[1] Johan C Fischer,[2] Janneke Harting,[3] Cees MPM Hertogh,[4] Jan M Prins,[5] Frank van Leth,[6] Menno D de Jong,[1] Caroline Schneeberger[1]

[1]Medical Microbiology, Amsterdam UMC University of Amsterdam, Amsterdam, The Netherlands
[2]Clinical Chemistry, Amsterdam UMC University of Amsterdam, Amsterdam, The Netherlands
[3]Public Health, Amsterdam UMC University of Amsterdam, Amsterdam, The Netherlands
[4]General Practice and Elderly Care Medicine, Amsterdam UMC Vrije Universiteit Amsterdam, Amsterdam, The Netherlands
[5]Internal Medicine, Amsterdam UMC University of Amsterdam, Amsterdam, The Netherlands
[6]Global Health, Amsterdam Institute for Global Health and Development, Amsterdam, The Netherlands

**Correspondence to**
Sacha D Kuil; s.d.kuil@amc.nl

## ABSTRACT

**Introduction** Suspected urinary tract infection (UTI) ranks among the most common reasons for antibiotic use in nursing homes. However, diagnosing UTI in this setting is challenging because UTI often presents with non-specific symptomatology. Moreover asymptomatic bacteriuria is common in elderly, which complicates attribution of causality to detection of bacteria in urine. These diagnostic challenges contribute to overuse of antibiotics and emergence of antimicrobial resistance in nursing homes. Given the diagnostic challenges, there is a need for point-of-care (POC) diagnostic tests to support clinical rules for diagnosing UTI. Procalcitonin (PCT) and C reactive protein (CRP) are inflammatory blood markers that have been proven useful to support diagnosis and monitoring of (bacterial) respiratory tract infections and sepsis. While limited studies suggest their usefulness in supporting UTI diagnosis, their utility has not been studied in elderly populations for this purpose.

**Methods and analysis** In a 24-month matched prospective study, 'PROGRESS' will assess and compare the sensitivity of rapid POC measurements of blood CRP and PCT levels to support clinical rules for diagnosing UTI in nursing home residents. The primary outcome measure is sensitivity of the POC tests to identify patients with true UTI based on the predefined definition, as derived from receiver operating curves.

**Ethics and dissemination** This study will be conducted in accordance with Good Clinical Practice guidelines and the principles of the Declaration of Helsinki. The study protocol is approved by the Medical Ethical Committee of Amsterdam UMC location VUmc with reference number 2017.350 and National Central Committee on Research involving Human Subjects with reference number NL62067.029.17.

**Trial registration number** NTR6467.

## Strengths and limitations of this study

- ► Stringent post hoc urinary tract infection criteria incorporating microbiology result and clinical response to adequate antibiotic therapy.
- ► Sensitivity in relevant study population, namely nursing home residents.
- ► No on-site point-of-care test (POCT), however, POCT is performed within 4 hours.
- ► Single country study.

in worldwide. Nursing homes are increasingly regarded as an important reservoir for the emergence of AMR.[1–4]

The most frequently reported infections in elderly residents are urinary tract infections (UTIs).[5] In Dutch nursing homes, an average weekly incidence of 10.3 (95% CI 9.8 to 10.8) per 1000 elderly residents was found.[6] UTI is the most common reason for prescribing antibiotics in Dutch nursing homes. However, not all diagnosed UTIs are 'true' UTIs since recognition and diagnosing UTI in nursing homes is complex. Approximately one-third of elderly residents UTIs are misdiagnosed, leading to inappropriate antibiotic use.[7 8] This is mainly due to the facts that non-specific, non-urinary tract symptoms, such as altered mental status, are often attributed to UTIs while asymptomatic bacteriuria (ASB), possibly resulting in positive urine tests, is also common in the elderly residents.[9]

### Cognitive impairments and UTIs

The majority of nursing homes in the Netherlands consist of psychogeriatric wards (57%), for elderly residents suffering from cognitive impairments, mainly Alzheimer disease.[10–13] Their ability to verbally

## INTRODUCTION

Antimicrobial resistance (AMR) is mainly driven by the inappropriate antibiotic use in both humans and animals. AMR is a problem

communicate or express classical symptoms of UTI, such as dysuria, urgency or frequency, is often limited.[14] The most frequently presented symptom in elderly residents leading to antibiotic prescription for a suspected UTI is an altered mental status (43.3%), while classical symptoms as dysuria, urgency and frequency are present in the minority of cases (0%–3.8%).[15] Confusion or an altered mental state is a non-specific symptom and can result from other infectious and non-infectious diseases in the elderly residents.

## Asymptomatic bacteriuria
ASB is defined by the presence of significant bacteriuria without symptoms of UTI. ASB is thus regarded as colonisation of the urinary tract rather than infection. ASB is highly prevalent in healthy elderly persons with reported prevalence rates as high as 40%–50%.[16 17] In the presence of ASB, frequently used urine tests based on detection of bacteria are less applicable to diagnose UTI, because detection of bacteria does not discriminate between ASB and UTI. In combination with above described non-specific symptomatology, it is difficult to distinguish ASB from 'true' UTI in the elderly.

## C reactive protein and procalcitonin
In the diagnosis of respiratory tract infections and monitoring of bacterial sepsis, inflammatory markers, such as C reactive protein (CRP) and procalcitonin (PCT) in blood, have proven useful to guide antibiotic therapy and reduce antibiotic use.[18–21] Currently, point-of-care (POC) CRP measurements are recommended by Dutch guidelines for the general practitioners to guide antibiotic treatment for acute respiratory tract infections. CRP and PCT represent potential candidates for rapid POC testing (POCT) to support UTI diagnosis.

Ageing and frailty are associated with changes in serum inflammation protein levels, such as CRP and PCT.[22–24] Therefore, it is important to determine cut-off values for CRP and PCT specifically in the elderly nursing home resident population.

## Inflammatory markers CRP and PCT in UTI
Studies in adults showed that CRP and PCT levels in parenchymatous infections (acute pyelonephritis, prostatitis and epididymitis) are increased.[25 26] Using a PCT-based algorithm in UTI treatment was shown to reduce antibiotic exposure in adults.[27] In a subgroup analysis including elderly (>70 years of age) with lower UTI antibiotic exposure was reduced as well, suggesting a potential role for PCT.

In children with UTI, sensitivity of CRP (cut-off 20 mg/L) and PCT (cut-off 0.5 ng/mL) in predicting pyelonephritis is high (94% and 86%, respectively), but specificity varies (39% and 74%, respectively)).[28] However, the number of studies in this systematic review was limited and the heterogeneity substantial. The specificity and positive predictive values varied because CRP and PCT were also increased in children with lower UTI.

This suggests a possible role for inflammatory markers in distinguishing UTI (upper and lower UTI) from no inflammation (ASB), where studies are lacking.

In the elderly population, only three studies have been performed evaluating CRP and PCT and their possible role in UTI.[29–31] These studies focused on hospitalised and more severely ill elderly populations, making the results not applicable to the nursing home population. In 13% of the healthy elderly controls, CRP values were increased,[30] again suggesting increased inflammation in ageing and frailty, indicating the need to determine specific cut-offs in elderly. Studies on lower UTIs and CRP or PCT in elderly are scarce.

## Distinguishing UTI and ASB
The majority of UTIs in elderly residents are lower UTIs without fever or other signs of systemic illness,[32 33] while most studies on CRP and PCT in UTI focus on diagnostic values in upper UTI or focus on distinguishing upper and lower UTI. The only small study on distinguishing lower UTI (infection) from ASB (colonisation) in adults, shows a high negative predictive value for UTI of PCT levels at a cut-off of 0.25 ng/mL[34], suggesting that low PCT levels can rule out UTI and contribute to reducing antibiotic use.

In conclusion, decisions about who to treat and who not to treat are challenging in elderly residents with suspected UTI, resulting in potential antibiotic overuse in those who do not need treatment. Availability of a simple and rapid POC test to distinguish true UTI from ASB represents an unmet need that would greatly assist clinical management of these vulnerable patients, by improving appropriateness of antibiotic treatment and by enabling consideration of alternative causes of presenting non-specific symptomatology when UTI is unlikely.

## METHODS AND ANALYSIS
### Aim
To assess the sensitivity of POC measurements of blood CRP and PCT levels to support clinical rules for diagnosing UTIs in elderly nursing home residents.

### Outcome
The primary outcome is the sensitivity of the POC test to identify patients with a true UTI based on the predefined definition, as derived from receiver operating curves (ROC).

### Design and setting
In a prospective matched study, we will assess and compare the sensitivity of rapid POC measurements of blood CRP and PCT levels to support clinical rules for diagnosing UTI in elderly residents, with a post hoc definition of UTI with stringent criteria including microbiology results as gold standard. The start date of this study is November 2017 and the planned end date is December 2019.

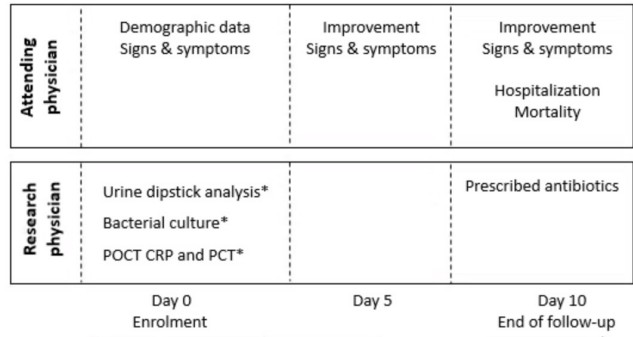

**Figure 1** Data collection progress study. *Participants and attending physicians/nurses will not be informed of results (blinded). CRP, C reactive protein; PCT, procalcitonin; POCT, point-of-care testing.

In this study, a true UTI is present when the following five criteria are met: presence of at least two urinary or non-specific symptoms (1) positive urine leucocyte esterase test (2) presence of uropathogens in bacterial culture at $10^4 \geq$ colony forming units (CFU)/mL (3) maximum of two uropathogens present (4) and symptom resolution in the course of adequate antibiotic treatment, where adequate treatment is defined by proven susceptibility of isolated uropathogens to the administered antibiotic (5).

The matching refers to the assessment of blood CRP and PCT levels in the same study participants. The study will be performed in nursing homes of the University Network for Organizations of Elderly Care of the VUmc University Medical Center. The expected study duration is 24 months.

The nursing home population in this study consists mostly of psychogeriatric and somatic (long-stay) wards and some rehabilitation (short-stay) wards.

### Informed consent procedure
Most nursing home patients are incapacitated. In case of incapacity legal, representatives will be asked for informed consent. Capacitated patients will be asked for informed consent themselves. When nursing home staff suspect an UTI, it is not considered practical to obtain written informed consent from the representatives because preferably the blood sample should be drawn as soon as possible. Therefore, informed consent will be obtained pre-emptively at the start of the study or when admitted to the nursing home. This means that patients or their legal representatives provide consent a priori to participate in the study once an UTI is clinically suspected during the study period. This procedure will greatly enhance feasibility of enrolment in psychogeriatric nursing home wards.

### Inclusion and exclusion criteria
Eligible for study participation is elderly nursing home residents clinically suspected of UTI by the attending physician or nurse. Exclusion criteria are suspected respiratory tract infection, suspected other infection requiring antibiotic therapy, previous study inclusion in the past 30 days or lack of written informed consent.

### Study procedures
#### Study enrolment
The study physician will be notified by the nursing home staff when there is potentially eligible patient and will visit the nursing home as soon as possible. The research physician will verify if eligibility criteria are met and will complete study enrolment.

#### Data collection
To meet our post hoc UTI criteria data on signs and symptoms, type and duration of antibiotic used, urine leucocyte esterase, bacterial culture and antimicrobial susceptibility and clinical response are collected. Demographic, clinical and laboratory data are collected through an electronic data capture system using software of Open Data Kit[35] with which case report forms (CRFs) are designed that incorporate consistency checks to minimise incompatible data points. Data will be handled encoded, working with barcodes scanned by the database APP. Data of paper registration forms are directly entered into online CRFs and uploaded to a predefined database.

Data on culture results (species, susceptibility patterns by minimal inhibitory concentrations) from the laboratory system will be collected in the currently used laboratory systems (Labtrain and Kiestra)

#### Data on signs and symptoms, clinical response and antibiotic use
The attending physician or nurse collects demographic data and data on signs and symptoms of UTI at the day of study enrolment (day 0), see figure 1. The attending physician or nurse will evaluate improvement of signs and symptoms at day 5 and 10 and hospitalisation and/or mortality at day 10. The research physician will visit the nursing homes regularly for monitoring of follow-up. Data on antibiotic use (timing of initial antimicrobial therapy, type of antimicrobial agent and possible switch in antimicrobial therapy) will be collected by the research physician 10 days after study enrolment.

#### Urine sample collection, dipstick analysis and bacterial culture
The attending nurse will collect an urine sample for purpose of this study. Urine samples are collected spontaneously voiding either directly in a sterile urine container or in chamber pot (insert pan). When participants have an indwelling urine catheter, urine will be collected when the urine bag is changed. When participants suffer from urinary incontinence or are not able to urinate on the toilet or chamber pot, diapers will be used to obtain urine for bacterial culture and dipstick analysis[36 37] Urine extraction from diapers can provide a reliable diagnostic specimen, if faecal contamination can reasonably excluded.[38] Date, time and way of urine collection method are registered by the attending nurse. The urine sample is stored at 4°C.

For dipstick analysis Combur 2 (Roche Diagnostics) will be used (nitrite and leucocyte esterase) by the research

physician. Urine will be used for semiquantitative bacterial culture. Inoculation of 10 µL urine to CHROMID CPS Elite (Chromogenic Medium for the immediate identification of *E. coli*, *Proteeae* and *Enterococcus*) agar (Biomerieux) and Columbia Colistin Nalidixin Acid (CNA) agar with 5% sheep blood (Biomerieux) will be streaked using a four quadrant pattern. Bacterial growth will be interpreted after overnight incubation at 35°C in aerobic (CPSE) and $CO_2$ enriched (CNA) environment. Uropathogens will be identified by Maldi-tof mass spectrometry (Microflex, Bruker Daltonic). All primary and secondary uropathogens in the European Consensus Guideline are considered as uropathogens in this study. Doubtful isolates are considered as non-uropathogens.[39] When bacterial growth of $\geq 10^4$ CFU/mL is found, antibiotic susceptibility testing will be performed using the VITEK2 platform (BioMérieux). Unlike the European Consensus Guideline, for all uropathogens $\geq 10^4$ CFU/mL growth will be used as cut-off, as urine collection methods will differ and the presence or absence of specific urinary symptoms. Participants, attending physicians and nurses will not be informed of urine dipstick and bacterial culture results (blinded).

### Blood sample collection and POCT
For POCT, the research physician will collect a blood sample by capillary fingerprick at day 0. Capillary blood sample collection is a preferred collection method in the elderly nursing home population. For CRP testing, the Afinion AS100 POC platform (Alere Health B.V., Tilburg) is used. For PCT testing, the Afias1 PCT Plus (Avant Medical B.V., Geffen) is used.

Both, CRP and PCT platforms enable capillary blood sample collection to better facilitate future implementation in nursing homes. POCT is performed according to the manufacturer's protocol. For logistic reasons, the POCT is performed centrally at the medical microbiology laboratory of our hospital. POCTs are performed within 4 hours after blood sample collection to ensure stability of CRP and PCT.

Participants, attending physicians and nurses will not be informed of POC test results (blinded).

### Biological specimen storage and molecular analysis in future
Aliquots of urine samples will be stored (−80°C) when consent is obtained specifically for genetic studies. In future studies metagenomic sequencing will be performed on bacterial DNA, to identify uropathogens and resistance genes.

### Sample size
For the sample size calculation, we used a two-sided p value of 0.05 and a power of 90%. We assume a difference in sensitivity between the two POCTs of at least 10% as clinically relevant. This difference is the net result of discordant test outcomes between the two tests. The magnitude of discordance is the main parameter in the McNemar test used to calculate the sample size for the matched design.

With a proposed sample size of 440 enrolled participants, we are able to adequately assess an increased sensitivity of 10% or more, when the prevalence of UTI in the study population is 40% or more. If the prevalence if UTI is lower, the difference in prevalence that is statistically significant (at the 5%) level increases to 11% or 12%.

Based on data of the national sentinel surveillance network for infectious diseases in nursing homes (SNIV) in 2015, the prevalence of UTI in Dutch nursing homes is 10.3 in 1000 patient weeks. UTI definition by SNIV is broader where response to adequate antibiotic treatment is not taken into account.

In order to reach the target, the PROGRESS study will take place in 13 nursing homes with around 1350 elderly residents in total. Recruitment of a manageable number of approximately 12 participants per week in total is anticipated.

### Data and statistical analysis
The sensitivity of both POCTs to diagnose UTIs defined by the post hoc definition is derived from the optimal point of an ROC. Sensitivities of CRP and PCT tests are compared by a matched analysis approach using a two-tailed McNemar test assessing the statistical significance of differences between the sensitivities of each pair of tests (performed in each study participant).

### Patient and public involvement
Patients or public are not involved in designing this study.

This study protocol is written using the Standard Protocol Items: Recommendations for Interventional Trials reporting guidelines.[40]

## DISSEMINATION
Written informed consent from nursing home residents or legal representatives (when incapacitated) will be obtained prior to study enrolment. Informed consent from residents will be obtained directly by researchers or via mail. Informed consent from legal representatives will be obtained via mail. Treating physicians decide which residents are capacitated.

### Data safety monitoring board
No data safety monitoring board has been appointed for this study as the risks of this study are assumed negligible.

## DISCUSSION
This trial will test the hypothesis that increased levels of inflammatory markers support the diagnosis of UTI in elderly and can guide empirical antibiotic treatment in nursing homes. Although our study involves the target population (elderly nursing home residents), found CRP and/or PCT cut-off values need to be confirmed by prospective trials. Besides test sensitivity, the potential impact on antibiotic prescription needs to be established

in a consecutive, randomised controlled, study.[41] The aim of this study is assessing the sensitivity of POCT CRP and PCT in current clinical practice. This clinical setting is suboptimal for urine collection, where urine incontinence complicates collection methods. Chamber pots and diapers are frequently used, which introduce the risk of UTI overestimation by positive dipstick urinalysis and bacterial cultures. Although we have shown in our add-on laboratory study that diapers can be used for UTI diagnosis, the outcome can be affected by suboptimal urine collection.

In this population, a reference test for UTIs does not exist, this is the actual gap we are trying to address. To reduce classification bias when a reference test is lacking, we use a post hoc definition of 'true' UTI.[42] We defined a stringent definition of UTI that in our opinion makes a clear distinction between UTI and ASB. We will perform post hoc analysis in which UTI definition is less stringent to assess the effects of potential misclassification of the outcome. UTI definition in this post hoc analysis will include classical UTI symptoms irrespective of resolution with adequate antibiotics, in accordance with the Dutch national guidelines.[43] We will include this procedure in the current amendment and data analysis plan that is drawn up before the completion of the data collection.

Since effectiveness of a new test does not just depend on its proven efficacy in research studies, but also on successful implementation after the study, qualitative research on identifying the barriers and facilitators for implementation are needed and will be performed in parallel to the described study.

## Protocol version

4 March 2019, v.5.

1 September 2017 original 21 October 2017 amendment 1: Addition of information brochures for nursing home staff and patients.

20 November 2017 amendment 2: Change of nomenclature for part of nursing home residents to temporary rehabilitation patients (at rehabilitation wards).

7 March 2018 amendment 3: Change exclusion criterion prior inclusion to prior inclusion in the past 30 days. Subsequently to account for non-independence of observation for a limited number of participants, the sample size is adjusted with a design effect of 1.1.

28 June 2018 amendment 4: (1) change in blood sample collection method (venipuncture to finger prick collection) due to availability of new PCT POCT with small input volume, as fingerprick blood sample collection is a preferred collection method in the elderly nursing home population, (2) change sample size retaining stringent criteria (p=value 0.05 and power 90%), (3) prolongation study duration: 12–18 months and (4) expansion of number of nursing homes from 11 to 12 nursing homes.

3 March 2019 amendment 5: (1) expansion of number of nursing homes from 12 to 13 nursing homes, (2) adjusted informed consent procedure for capacitated residents.

1 July 2019 amendment 6: (1) expansion of number of nursing homes from 13 to 14 nursing homes and (2) addition of post hoc analysis using different UTI definitions.

**Contributors** SDK recruited patients, coordinates and performed the clinical study and the wrote the manuscript. SH recruited patients and performed the clinical study. FvL, JCF, JH, CMH, JP, MDdJ and CS developed the protocol and secured funding for this project. CS, FvL and MDdJ supervised the design of the study and writing this manuscript. FvL provided the statistical analysis plan, database set-up. All authors have read and approved the manuscript. Research physician/PhD student (SDK): coordinating PROGRESS, patient recruiting, data collection, specimen handling, POC and urine testing, preparation protocols, CRFs and publication of study reports including annual ethical committee report. Research assistant (SH): patient recruiting, data collection, specimen handling, POC and urine testing. Principal investigator (MDdJ), senior researcher (CS) and lead epidemiologist (FvL): Study planning, agreement of final protocols, reviewing progress of study. Lead epidemiologist (FvL): design of the electronic report form (eCRF), database design and maintenance, data validation and data management plans, user account maintenance, design of automated data validation scripts, preparation of the final data anlysis sets, supervising of data analysis. Laboratory chemist (JCF): responsible for verification of POCT, contractual issues with manufacturers. In each participating center, a lead investigator (elderly care physician) will be identified, to be responsible for progress monitoring and assisting with study set-up per site.

**Funding** This study is funded by The Netherlands Organization for Health Research and Development (ZonMW) grant no 541001003. ZonMW, Laan van Nieuw Oost Indië 334, 2593 CE Den Haag, The Netherlands. The study protocol is being peer reviewed by the funder.

**Disclaimer** According to the CCMO statement on publication policy, the results of this study will be disclosed unreservedly. The funder has no role in collection, management, analysis, and interpretation of data, writing of the report or the decision to submit the report for publication.

**Competing interests** None declared.

**Patient consent for publication** Not required.

**Ethics approval** This study will be conducted in accordance with Good Clinical Practice guidelines and the principles of the Declaration of Helsinki. The study protocol is approved by the Medical Ethical Committee (METc) of Amsterdam UMC location VUmc with reference number 2017.350 and National Central Committee on Research involving Human Subjects (CCMO) with reference number NL62067.029.17.

**Provenance and peer review** Not commissioned; externally peer reviewed.

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
