## [Reviewer comments · BMJ Open]

ARTICLE DETAILS

TITLE (PROVISIONAL)	Sensitivity of point of care testing C-reactive protein and procalcitonin to diagnose urinary tract infections in Dutch nursing homes: PROGRESS study protocol
AUTHORS	Kuil, Sacha; Hidad, Soemeja; Fischer, Johan; Harting, Janneke; Hertogh, C; Prins, Jan; van Leth, Frank; de Jong, Menno; Schneeberger, Caroline

VERSION 1 – REVIEW

REVIEWER	Michal Nowicki Medical University of Lodz, Poland
REVIEW RETURNED	16-May-2019

GENERAL COMMENTS	There is a need for a study that will assess the use of point of care testing of UTI in a population at high risk of the disease. The study is well-design and important from the practical and organizational point of view.
---

REVIEWER	Anne Holm University of Copenhagen
REVIEW RETURNED	21-May-2019

GENERAL COMMENTS	Thank you for a well-planned and very relevant study. Antibiotic overuse in the elderly is indeed an increasing problems and research into solutions to this problem is needed. I have some corrections and suggestions below, which should be looked into before the protocol is ready for publication. My main concern is how you address your aim: The aim of your study is to evaluate the utility of POC tests, but your study is a classic diagnostic accuracy study not evaluating utility 1. The other limitation of the study is how you address the lack of a clear reference. This is indeed challenging since there is no clear definition of UTI in this population to distinguish those who recover without therapy from those in need of antibiotics. Besides your already chosen reference, I would recommend, you look into statistical methods, which can be used in lack of a clear reference standard and also compare with other definitions in use 2,3. Your aim is to assess the utility of POC tests, but in your statistical analysis you aim to evaluate sensitivity only? This is not a very clinically relevant endpoint or very relevant in order to reduce antibiotic overuse. Since you have chosen to perform a classic diagnostic study although no reference standard exists, it seems more appropriate to calculate likelihood ratios as well as pre- and post-test probabilities. Sensitivity is likely to be high while specificity will be low due to reference standard misclassification 5.
--

	Besides this I have some minor comments: Your definition of significant growth is 104 cfu/ml (regardless the uropathogen). This is not according to European consensus 4. Any reason for this? It is challenging to obtain a clean urine sample in nursing homes as you are clearly aware. I would suggest you take different sampling techniques into account in your statistical analysis since contamination will compromise your reference standard. Otherwise you may miss relevant differences between the groups of sampling techniques As you explain, inflammation markers are elevated in the elderly population and associated to frailty. Since your population may be more or less frail than others, baseline values of crp and procalcitonin would increase external validity of your findings. Since the study is ongoing, this may not be applicable. You do not have a section on blinding (of the reader of index tests and reference as well as the one registering clinical information). A figure of your data collection could be helpful. It is not completely clear how clinical data is registered and when. You may want to add to your conclusions that since you do not have a predefined cut-off of your POC tests, your results will have to be confirmed by prospective trials using your cut-offs. You need to discuss limitations of the study more thoroughly in the discussion. A few references that may be relevant in your introduction 6–8. Suggested references  1. Bossuyt PMM, Reitsma JB, Linnet K, Moons KGM. Beyond Diagnostic Accuracy : The Clinical Utility of Diagnostic Tests. Clin Chem. 2012;58(12):1636-1643. doi:10.1373/clinchem.2012.182576. 2. Gunnarsson RK, Lanke J. The predictive value of microbiologic diagnostic tests if asymptomatic carriers are present. Stat Med. 2002;21:1773-1785. doi:10.1002/sim.1119. 3. Nace DA, Drinka PJ, Crnich CJ. Clinical Uncertainties in the Approach to Long Term Care Residents With Possible Urinary Tract Infection. J Am Med Dir Assoc. 2014;15(2):133-139. 4. Aspevall O, Hallander H, Gant V, Kouri T. European guidelines for urinalysis: a collaborative document produced by European clinical microbiologists and clinical chemists under ECLM in collaboration with ESCMID. Scand J Clin Lab Invest. 2000;60:1-96. 5. Biesheuvel C, Irwig L, Bossuyt P. Observed differences in diagnostic test accuracy between patient subgroups: Is it real or due to reference standard misclassification? Clin Chem. 2007;53(10):1725-1729. 6. Shaikh N, Borrell JL, Evron J, Leeflang MMG. Procalcitonin, C-reactive protein, and erythrocyte sedimentation rate for the diagnosis of acute pyelonephritis in children. Cochrane database Syst Rev. 2015;(1):CD009185. doi:10.1002/14651858.CD009185.pub2. 7. Aabenhus R, Jensen JUS, Jørgensen KJ uhl, Hróbjartsson A, Bjerrum L. Biomarkers as point-of-care tests to guide prescription of antibiotics in patients with acute respiratory infections in primary care. Cochrane database Syst Rev. 2014;11. doi:10.1002/14651858.CD010130.pub2. 8. Schuetz P, Wirz Y, Sager R, et al. Procalcitonin to initiate or discontinue antibiotics in acuterespiratory tract infections (Review). Cochrane Database Syst Rev. 2017;Art. No.:(10). doi:10.1002/14651858.CD007498.pub3.www.cochranelibrary.com.
--	---

REVIEWER	Per Åkesson Department of Infectious Diseases Skane University Hospital Lund University Lund Sweden I am a co-author on a application for Heparin-binding protein (HBP) as a biomarker for infection. The patent is owned by Hansa Biopharma, Lund, Sweden, and licenced to Axis-Shield, Dundee, UK.
REVIEW RETURNED	28-May-2019

GENERAL COMMENTS	From my veiwpoint, the planned study is well motivated with sound methodology and suggested outcome. I have two personal suggestions that in my eyes would improve the study and make future results easier to trust.  1. Page, line 208 and 224: Researchers plan to enroll incapacitated subjects and at the same time use non-specific symptoms as a criteria for infection. This will inevitably lead to misclassification of study patients. 2. Page 10, line 278: Researchers plan to use urine from diapers and collect urine from bags when changed. Even if a reference on this procedure is provided, few readers will rule out contamination and subsequent misclasification of patients. Instead a syringe should be used for catheters and the diapers should not be used. Correct classification of patients is crucial in the described study. To differ ABU from true infection is difficult. To get reliable data, authors should aim for quality even if the study cohort will be smaller.
---

VERSION 1 – AUTHOR RESPONSE

REVIEWER 1

Reviewer Name: Michal Nowicki

Institution and Country: Medical University of Lodz, Poland

Please state any competing interests or state 'None declared': none declared

Please leave your comments for the authors below

There is a need for a study that will assess the use of point of care testing of UTI in a population at high risk of the disease. The study is well-design and important from the practical and organizational ponit of view.

We would like to thank the reviewer for the supporting comments.

REVIEWER 2

Reviewer Name: Anne Holm

Institution and Country: University of Copenhagen

Please state any competing interests or state 'None declared': None declared

Please leave your comments for the authors below

Thank you for a well-planned and very relevant study. Antibiotic overuse in the elderly is indeed an increasing problem and research into solutions to this problem is needed. I have some corrections and suggestions below, which should be looked into before the protocol is ready for publication.

MAJOR COMMENTS

1) My main concern is how you address your aim: The aim of your study is to evaluate the utility of POC tests, but your study is a classic diagnostic accuracy study not evaluating utility 1.

As the reviewer rightly commented this is not a utility study but a diagnostic accuracy study evaluating the sensitivity of CRP and PCT measured using point-of-care tests to support clinical rules for diagnosing urinary tract infections (UTI) in elderly nursing home residents.

Therefore we replaced the aim in the methods section page 7 line 208-210:

“To assess the sensitivity of point-of-care measurements of blood CRP and PCT levels to support clinical rules for diagnosing urinary tract infections (UTI) in elderly nursing home residents.”

A study to evaluate the utility of POCT will be planned if sensitivity of $\geq 65\%$ in either or both of the tests has been shown in the current study. The test with the highest sensitivity will be selected for a consecutive study. If two tests show a sensitivity above 65%, then we formally test whether the higher sensitivity is statistically significant from the lower sensitivity, by applying a two-sided hypothesis test. This consecutive study will be a cluster randomized trial to assess the impact of POC CRP or PCT testing on antibiotic use in nursing home residents with suspected UTI. The primary outcome will be the total duration of antibiotic exposure during 30 days following study enrolment.

2) The other limitation of the study is how you address the lack of a clear reference. This is indeed challenging since there is no clear definition of UTI in this population to distinguish those who recover without therapy from those in need of antibiotics. Besides your already chosen reference, I would recommend you look into statistical methods, which can be used in lack of a clear reference standard and also compare with other definitions in use 2,3.

We will perform post-hoc analysis to assess the effects of potential misclassification of our definition, by using less stringent criteria for UTIs. Therefore we added a sentence in the discussion section, page 12-13, lines 406-414:

“In this population a reference test for UTIs does not exist, this is the actual gap we are trying to address. To reduce classification bias when a reference test is lacking we use a post-hoc definition of ‘true’ UTI. We defined a stringent definition of UTI that in our opinion makes a clear distinction between UTI and ASB. We will perform post-hoc analysis in which UTI definition is less stringent to assess the effects of potential misclassification of the outcome. UTI definition in this post-hoc analysis will include classical UTI symptoms irrespective of resolution with adequate antibiotics, in accordance with the Dutch national guidelines. We will include this procedure in the current amendment and data analysis plan that is drawn up before the completion of the data collection.”

3) Your aim is to assess the utility of POC tests, but in your statistical analysis you aim to evaluate sensitivity only? This is not a very clinically relevant endpoint or very relevant in order to reduce antibiotic overuse. Since you have chosen to perform a classic diagnostic study although no reference

standard exists, it seems more appropriate to calculate likelihood ratios as well as pre- and post-test probabilities. Sensitivity is likely to be high while specificity will be low due to reference standard misclassification 5.

As mentioned in comment 1 this study is a matched diagnostic accuracy study to assess sensitivities of the two proposed markers for the diagnosis of UTI in elderly nursing home residents. Although no reference test for UTIs in elderly exist, a stringent post-hoc definition for UTIs will be used to minimize misclassification. The primary outcome of this study is not reduction in antibiotic use, but the sensitivity of the POCT, to identify patients with true UTIs. Empirical estimated POCT cut-off levels will be derived from ROC analysis. Sensitivities will be calculated at estimated cut-off levels. Clinical utility will be addressed in a next study, once sensitivity of $\geq 65\%$ in either or both of the test has been shown in the current study (see comment 1).

Besides this I have some minor comments:

MINOR COMMENTS

1) Your definition of significant growth is 104 cfu/ml (regardless the uropathogen). This is not according to European consensus 4. Any reason for this?

Our definition of significant growth is ≥ 104 CFU/mL growth of a uropathogen. We decided to use primary and secondary uropathogens listed in the European Consensus Guideline. In this guideline however, different CFU/mL cutoff values are used depending on the urine sample type and identified uropathogen and presence of symptoms. In our study population we include various urine sample types and the main urine sample type (midstream urine) is hard to obtain, besides the presence of urinary specific symptoms are difficult to recognize (as stated in our rationale). Therefore we have chosen for an stringent cut-off of 104 CFU/mL.

We have added a few sentences on defined uropathogens and cut-off level in method section,

1. Page 10 line 307-309:

“All primary and secondary uropathogens in the European Consensus Guideline are considered as uropathogens in this study. Doubtful isolates are considered as non-uropathogens.”

2. Page 10 lines 310-312:

“Unlike the European Consensus Guideline, for all uropathogens ≥ 104 CFU/mL growth will be used as cut-off, as urine collection methods will differ and the presence or absence of specific urinary symptoms.”

2) It is challenging to obtain a clean urine sample in nursing homes as you are clearly aware. I would suggest you take different sampling techniques into account in your statistical analysis since contamination will compromise your reference standard. Otherwise you may miss relevant differences between the groups of sampling techniques

Obtaining a proper urine sample in nursing homes is definitely challenging and therefore we performed a pilot study “Diaper urine: a reliable alternative for obtaining urine samples for UTI diagnosis in elderly suffering from urine incontinence. Poster session P2137, 24 April 2018 Session: Urinary tract infection - risks, diagnosis, treatment. ECCMID 2018 Madrid, Spain” (manuscript in preparation).

In this study (252 diapers) we showed that diaper urine extraction was not inferior to direct use of urine for diagnosis of urinary tract infections by dipstick analysis and bacterial culture.

We expanded the discussion on diaper urine collection, page 12, lines 395-399:

“This clinical setting suboptimal for urine collection, where urine incontinence complicates collection methods. Chamber pots and diapers are frequently used, which introduces the risk of UTI overestimation by positive dipstick urinalysis and bacterial cultures. Although we have shown in our add-on laboratory study that diapers can be used for UTI diagnosis, the outcome can be affected by suboptimal urine collection.”

We register the way the urine sample are collected (collected spontaneously voiding either directly in a sterile urine container or in chamber pot (insert pan), but our sample size is not sufficient to take collection techniques into account in statistical analysis.

3) As you explain, inflammation markers are elevated in the elderly population and associated to frailty. Since your population may be more or less frail than others, baseline values of crp and procalcitonin would increase external validity of your findings. Since the study is ongoing, this may not be applicable.

We only collect CRP and PCT POCT test results when UTI is suspected by the treating physician. Therefore we cannot compare differences in POCT results from baseline.

4) You do not have a section on blinding (of the reader of index tests and reference as well as the one registering clinical information).

The clinical information will be assessed and registered by the treating physician. The treating physician does not have access to the results of the CRP or PCT POCT, urine dipstick or bacterial culture results, as stated at page 10 lines 331-332. We added an additional sentence on this subject in the methods page 10 lines 316-318 and a flow chart (Figure 1) to make this more clear.

“Participants, attending physicians and nurses will not be informed of urine dipstick and bacterial culture results (blinded).”

5) A figure of your data collection could be helpful. It is not completely clear how clinical data is registered and when.

We have added a figure on data collection (Figure 1), methods section, page 9.

6) You may want to add to your conclusions that since you do not have a predefined cut-off of your POC tests, your results will have to be confirmed by prospective trials using your cut-offs.

We added an additional sentence in the discussion section, page 12, line 396-398:

“Although our study involves the target population (elderly nursing home residents), found CRP and/or PCT cut-off values need to be confirmed by prospective trials.”

7) You need to discuss limitations of the study more thoroughly in the discussion.

As the reviewer suggested we expanded the discussion and elaborated on the limitations: confirmation of CRP/PCT cut-off levels in prospective trial, lack of reference test and urine collection methods (page 12-13, lines 396-414)

“Although our study involves the target population (elderly nursing home residents), found CRP and/or PCT cut-off values need to be confirmed by prospective trials. Besides test sensitivity the potential impact on antibiotic prescription needs to be established in a consecutive, randomized controlled, study. The aim of this study is assessing the sensitivity of POCT CRP and PCT in current clinical practice. This clinical setting suboptimal for urine collection, where urine incontinence complicates collection methods. Chamber pots and diapers are frequently used, which introduces the risk of UTI overestimation by positive dipstick urinalysis and bacterial cultures. Although we have shown in our add-on laboratory study that diapers can be used for UTI diagnosis, the outcome can be affected by suboptimal urine collection.

In this population a reference test for UTIs does not exist, this is the actual gap we are trying to address. To reduce classification bias when a reference test is lacking we use a post-hoc definition of ‘true’ UTI. We defined a stringent definition of UTI that in our opinion makes a clear distinction between UTI and ASB. We will perform post-hoc analysis in which UTI definition is less stringent to assess the effects of potential misclassification of the outcome. UTI definition in this post-hoc analysis will include classical UTI symptoms irrespective of resolution with adequate antibiotics, in accordance with the Dutch national guidelines. We will include this procedure in the current amendment and data analysis plan that is drawn up before the completion of the data collection.”

A few references that may be relevant in your introduction 6–8.

Suggested references

1. Bossuyt PMM, Reitsma JB, Linnet K, Moons KGM. Beyond Diagnostic Accuracy : The Clinical Utility of Diagnostic Tests. *Clin Chem*. 2012;58(12):1636-1643. doi:10.1373/clinchem.2012.182576.
2. Gunnarsson RK, Lanke J. The predictive value of microbiologic diagnostic tests if asymptomatic carriers are present. *Stat Med*. 2002;21:1773-1785. doi:10.1002/sim.1119.
3. Nace DA, Drinka PJ, Crnich CJ. Clinical Uncertainties in the Approach to Long Term Care Residents With Possible Urinary Tract Infection. *J Am Med Dir Assoc*. 2014;15(2):133-139.
4. Aspevall O, Hallander H, Gant V, Kouri T. European guidelines for urinalysis: a collaborative document produced by European clinical microbiologists and clinical chemists under ECLM in collaboration with ESCMID. *Scand J Clin Lab Invest*. 2000;60:1-96.
5. Biesheuvel C, Irwig L, Bossuyt P. Observed differences in diagnostic test accuracy between patient subgroups: Is it real or due to reference standard misclassification? *Clin Chem*. 2007;53(10):1725-1729.
6. Shaikh N, Borrell JL, Evron J, Leeflang MMG. Procalcitonin, C-reactive protein, and erythrocyte sedimentation rate for the diagnosis of acute pyelonephritis in children. *Cochrane database Syst Rev*. 2015;(1):CD009185. doi:10.1002/14651858.CD009185.pub2.
7. Aabenhus R, Jensen JUS, Jørgensen KJ uhl, Hróbjartsson A, Bjerrum L. Biomarkers as point-of-care tests to guide prescription of antibiotics in patients with acute respiratory infections in primary care. *Cochrane database Syst Rev*. 2014;11. doi:10.1002/14651858.CD010130.pub2.
8. Schuetz P, Wirz Y, Sager R, et al. Procalcitonin to initiate or discontinue antibiotics in acuterespiratory tract infections (Review). *Cochrane Database Syst Rev*. 2017;Art. No.:(10). doi:10.1002/14651858.CD007498.pub3. www.cochranelibrary.com.

We would like to thank the reviewer for the additional references. We have added the references 4 in our methods section, 1 in discussion section and 6-8 in our introduction section.

REVIEWER 3

Reviewer Name: Per Åkesson

Institution and Country: Department of Infectious Diseases, Skane University Hospital, Lund University, Lund, Sweden

Please state any competing interests or state 'None declared': I am a co-author on a application for Heparin-binding protein (HBP) as a biomarker for infection. The patent is owned by Hansa Biopharma, Lund, Sweden, and licenced to Axis-Shield, Dundee, UK.

Please leave your comments for the authors below

From my veiwpoint, the planned study is well motivated with sound methodology and suggested outcome. I have two personal suggestions that in my eyes would improve the study and make future results easier to trust.

1) Page, line 208 and 224: Researchers plan to enroll incapacitated subjects and at the same time use non-specific symptoms as a criteria for infection. This will inevitably lead to misclassification of study patients.

The debate about the non-specific symptoms is one of the main reasons for this study. With the stringent UTI definition proposed in the protocol where we also included the clinical response (improvement of the patient, reduction of the symptoms including non-specific symptoms) we aim to reduce misclassification of the outcome. We will perform post-hoc analysis to assess the effects of potential misclassification of our definition, by using less stringent criteria for UTIs. Therefore we added a sentence in the discussion section, page 12-13, lines 406-414:

“In this population a reference test for UTIs does not exist, this is the actual gap we are trying to address. To reduce classification bias when a reference test is lacking we use a post-hoc definition of 'true' UTI . We defined a stringent definition of UTI that in our opinion makes a clear distinction between UTI and ASB. We will perform post-hoc analysis in which UTI definition is less stringent to assess the effects of potential misclassification of the outcome. UTI definition in this post-hoc analysis will include classical UTI symptoms irrespective of resolution with adequate antibiotics, in accordance with the Dutch national guidelines . We will include this procedure in the current amendment and data analysis plan that is drawn up before the completion of the data collection.”

2) Page 10, line 278: Researchers plan to use urine from diapers and collect urine from bags when changed. Even if a reference on this procedure is provided, few readers will rule out contamination and subsequent misclassification of patients. Instead a syringe should be used for catheters and the diapers should not be used.

Correct classification of patients is crucial in the described study. To differ ABU from true infection is difficult. To get reliable data, authors should aim for quality even if the study cohort will be smaller.

We understand the concern of the reviewer, however the aim of this study is to assess POCT sensitivity in current clinical practice in nursing homes. In this clinical setting urine collection methods are suboptimal and it is not feasible to create an optimal study setting. We assessed if diapers can be used in UTI diagnosis by bacterial culture and urine dipstick analysis. In a non-inferiority laboratory study we have compared urine bacterial culture and dipstick analysis results in over 250 urine samples. This study showed non-inferiority in UTI diagnosis by urine extracted from diapers.

We have extended the discussion with the following sentences, page 12 lines 399-405

“The aim of this study is assessing the sensitivity of POCT CRP and PCT in current clinical practice. This clinical setting suboptimal for urine collection, where urine incontinence complicates collection methods. Chamber pots and diapers are frequently used, which introduces the risk of UTI overestimation by positive dipstick urinalysis and bacterial cultures. Although we have shown in our add-on laboratory study that diapers can be used for UTI diagnosis, the outcome can be affected by suboptimal urine collection.”

Reitsma JB, Rutjes AWS, Khan KS, Coomarasamy A, Bossuyt PM. A review of solutions for diagnostic accuracy studies with an imperfect or missing reference standard. *Journal of Clinical Epidemiology*. 2009; 62:797e806

Verenso. Richtlijn Urineweginfecties bij kwetsbare ouderen. 2018. Dutch national guideline urinary tract infections in elderly.

Bossuyt PMM, Reitsma JB, Linnert K, Moons KGM. Beyond Diagnostic Accuracy : The Clinical Utility of Diagnostic Tests. *Clin Chem*. 2012;58(12):1636-1643. doi:10.1373/clinchem.2012.182576.

Reitsma JB, Rutjes AWS, Khan KS, Coomarasamy A, Bossuyt PM. A review of solutions for diagnostic accuracy studies with an imperfect or missing reference standard. *Journal of Clinical Epidemiology*. 2009; 62:797e806

Verenso. Richtlijn Urineweginfecties bij kwetsbare ouderen. 2018. Dutch national guideline urinary tract infections in elderly.

Reitsma JB, Rutjes AWS, Khan KS, Coomarasamy A, Bossuyt PM. A review of solutions for diagnostic accuracy studies with an imperfect or missing reference standard. *Journal of Clinical Epidemiology*. 2009; 62:797e806

Verenso. Richtlijn Urineweginfecties bij kwetsbare ouderen. 2018. Dutch national guideline urinary tract infections in elderly.

VERSION 2 – REVIEW

REVIEWER	Anne Holm University of Copenhagen
REVIEW RETURNED	15-Jul-2019
GENERAL COMMENTS	Thank you for the revised manuscript, which has substantially improved.